

# Technical note: Problem specific variators in a genetic algorithm for the optimization of drinking water networks

Karel van Laarhoven[1], Ina Vertommen[1], Peter van Thienen[1]

[1]KWR Watercycle Research Institute, Nieuwegein, 3430 BB, the Netherlands

*Correspondence to*: Karel van Laarhoven (karel.van.laarhoven@kwrwater.nl)

**Abstract.** Genetic algorithms can be a powerful tool for the automated design of optimal drinking water distribution networks. Fast convergence of such algorithms is a crucial factor for successful practical implementation at the drinking water utility level. In this technical note, we therefore investigate the performance of a suite of genetic variators that was tailored to the optimisation of a least-cost network design. Different combinations of the variators are tested in terms of convergence rate and

the robustness of the results during optimisation of the real world drinking water distribution network of Sittard, the Netherlands. The variator configurations that reproducibly reach the furthest convergence after $10^5$ function evaluations are reported. In the future these may aid in dealing with the computational challenges of optimizing real world networks.

## 1 Introduction

Optimization techniques have been applied to the design (or more specifically, the dimensioning) of water networks for

decades (see Bieupoude et al., 2012 and De Corte and Sörensen, 2013, for overviews). A widely applied approach is that of Genetic Algorithms (GA) (Holland, 1975; Goldberg, 1989). Though the classic genetic algorithm is very powerful, a number of approaches have been developed to improve convergence in water network design problems. More specifically, the various mechanisms of the genetic algorithm are commonly expanded, replaced or combined with heuristic tricks or complete heuristic algorithms. The algorithms which include these are commonly referred to as Hybrid Genetic Algorithms (HGA) or Memetic

Algorithms (MA). An overview of these approaches is summarized below. Following one of the approaches, a selection of custom heuristic variators has been implemented in Gondwana, a generic optimization tool for drinking water networks (Van Thienen and Vertommen, 2015). In this paper, we describe these variators and demonstrate how they contribute to significantly faster convergence in a sample optimization problem.

## 2 Hybrid Genetic Algorithms

A cornerstone of the HGA approach (Krasnogor and Smith, 2005; El-Mihoub et al., 2006) is the observation that classic GA are especially well suited for quickly locating global optima in the solution space, but subsequently have difficulty converging to the optimum locally within a reasonable number of iterations. To mitigate this, GA are augmented with Local Search (LS) methods. These are algorithms that iteratively modify a given solution towards a predefined optimization criterion. LS methods find local optima relatively quickly, but are generally unable to escape this local optimum in favor of a possibly different global





optimum. The resulting HGA therefore profits from the strengths of both techniques and yields better solutions. El-Mihoub et al. identify the following general ways in which GA capabilities can be expanded through hybridization:

1. A GA solution can be improved by running it through a LS method. This can be done to improve the final solution of the GA. Alternatively, the LS algorithm can be applied to refine intermediate solutions to promote the representation of different promising areas of the solution space within the population.

2. The number of iterations that are needed to achieve convergence can be reduced by replacing classic genetic operators with different ones to guide the search through the solution space.

3. Alternatively, system specific knowledge can be used to modify the genetic operators in such a way that they only result in viable solutions. This does not guide the search, but prevents time loss due to the evaluation of many illegal solutions, which may arise from random variations in heavily restricted GA problems.

4. The population size needed to achieve convergence can be reduced by dynamically controlling candidate selection with a LS method.

5. System specific knowledge can be used to construct an model to quickly approximate the results of fitness functions that are expensive to calculate, speeding up the evaluation of the GA objectives.

## 3 Problem specific variators

Within the field of water network design optimization, algorithms that guide the GA to reduce the size of the search space is a specific challenge in current research (Maier et al., 2014). Table 1 lists a collection of genetic operators that was composed to tune a GA to the optimization of a least cost design (Alperovitz and Shamir, 1977; Savic and Walters, 1997). This type of problem varies pipe diameters throughout the network in search of the minimum network costs while achieving a minimum pressure at each node. In addition to several classic GA variators, two heuristic variators are used that were constructed with the goal of a least cost design in mind. In terms of the classification of hybrid metaheuristics by (Talbi, 2002), the resulting HGA is a low-level teamwork hybrid.

The heuristic flatiron mutator enhances convergence according to approach 2 in the list above. It guides the search past a type of artefact that commonly occurs in intermediate solutions for the least cost design problem. This artefact occurs when classic mutation causes a larger diameter pipe to be surrounded by smaller diameter pipes, which is hydraulically insensible. These artefacts can take a long time to disappear through random mutation only. The flatiron mutator speeds up convergence by detecting and 'smoothing out' these artefacts every iteration.

The heuristic list proximity mutator enhances convergence according to approach 3 in the list above. It is equivalent to the classic 'creep mutator' (Sivanandam & Deepa, 2007). It limits the possible outcomes of a mutation to diameters close to the original value, because large deviations from the original diameter are likely to cause hydraulically inviable solutions.



**Tests**

In order to evaluate the influence of the developed problem specific variators, a series of tests was performed on a case study. The case study consists on the design of part of the existing drinking water distribution network of the Dutch village Sittard. The network has a total length of 10.8 km and has 1000 connections, including connections to a school, a residential building

with 32 apartments and a care farm for mental patients. The network is fed by a single reservoir and has a mean total demand of 15 m3/h. The network is represented by an EPANET model consisting of 584 nodes, 491 and 1 reservoir. The network is displayed in Fig. 1.

For the design of the network the minimization of the product between pipe diameter and pipe length (surrogate for costs) was considered as the objective, constrained by a minimum pressure at each node equal to 34 meters. The decision variables were

10 the pipe diameters, that could be chosen from the set of available diameters summarized in Table 2.

A total of 16 tests were performed, wherein different values for the specific variators were considered in order to assess their influence on the network design results. For each test a total of 1x105 function evaluations and 10 runs were performed, i.e., each test was repeated 10 times in order to assess the mean, standard deviation, best and worst results obtained for each test. Table 3 provides an overview of the different tests, including the considered variator values and obtained results. Tests 1 and

15 2 consist on different value combinations of the classical naïve random mutation (RM) and one-point crossover (NPC). The heuristic proximity mutation was added in tests 3 to 8. Different values for the heuristic flatiron mutation were considered in tests 9 to 14. Tests 15 and 16 further explore the influence of the one-point crossover on the performance of the algorithm.

**Results**

From the obtained results (Table 3) it is clear that the consideration of the heuristic flatiron mutation (FM) and proximity

mutation (LPM) significantly improved the obtained results for the optimization problem. These results are graphically reported in Fig. 2.

Considering only the naïve random mutation and one-point crossover, the best results after 1x105 function evaluations were achieved with a mutation rate equal to 0.05 and a crossover rate equal to 0.95. In this case the average objective function value was 8.8x105. Adding a proximity mutation does not improve the results, but considering only a proximity mutation and no

random mutation, has a significant influence on the outcomes. Considering only the proximity mutation, the best results after 1x105 function evaluations were achieved for a proximity mutation rate of 0.05. With this value the best results were achieved for both the mean, best and worst values for the objective function. Fig. 3a illustrates the influence of this variator on the computed objective function values.

Adding a flatiron mutation further improved the obtained results. The best results after 1x105 function evaluations, on average,

were obtained for a combination of a crossover rate of 0.95, with a proximity mutation rate equal to 0.05 and a flatiron mutation rate equal to 0.8. The best result in one test only was obtained for slightly lower flatiron mutation rate, equal to 0.7. Fig. 3b illustrates the influence of this flatiron mutation rate on the obtained objective function values.





The effect of the problem specific variators can also clearly be seen on the shape of the convergence curves. Fig. 4 illustrates the mean, mean ± standard deviation, and mean ± 2 × standard deviation convergence curves for tests number 2 (RM-NPC2) and 12 (FM4). The proximity and flatiron mutations lead to smoother curves and a faster convergence. The standard deviation between results of the different runs is also much lower, which means that the results are more stable.

## 5  Discussion and conclusions

The results presented in this paper clearly illustrate the value of applying heuristic, non-classical variators in drinking water distribution system design optimizations using genetic algorithms.

In our tests, the combination of a low rate for the proximity mutation with a high rate of the flatiron mutation leads to the best results after $1 \times 10^5$ function evaluations (test numbers 13:FM5 and 14:FM6), i.e. the fastest convergence. All tested combinations which include either the flatiron or the proximity mutation exhibit a similar or worse performance. Albeit slower, particularly stable results were obtained with the proximity mutation (rate=0.1) and no flatiron mutation. These runs show the smallest standard deviation in the results after $1 \times 10^5$ function evaluations.

In our research and consulting projects, we will continue to use this combination of variators in order to deal with the computational challenges of larger real world networks.

## Acknowledgments

The authors wish to thank Henk Vogelaar from Waterleiding Maatschappij Limburg (WML) for providing the Sittard network model used in the calculations.

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

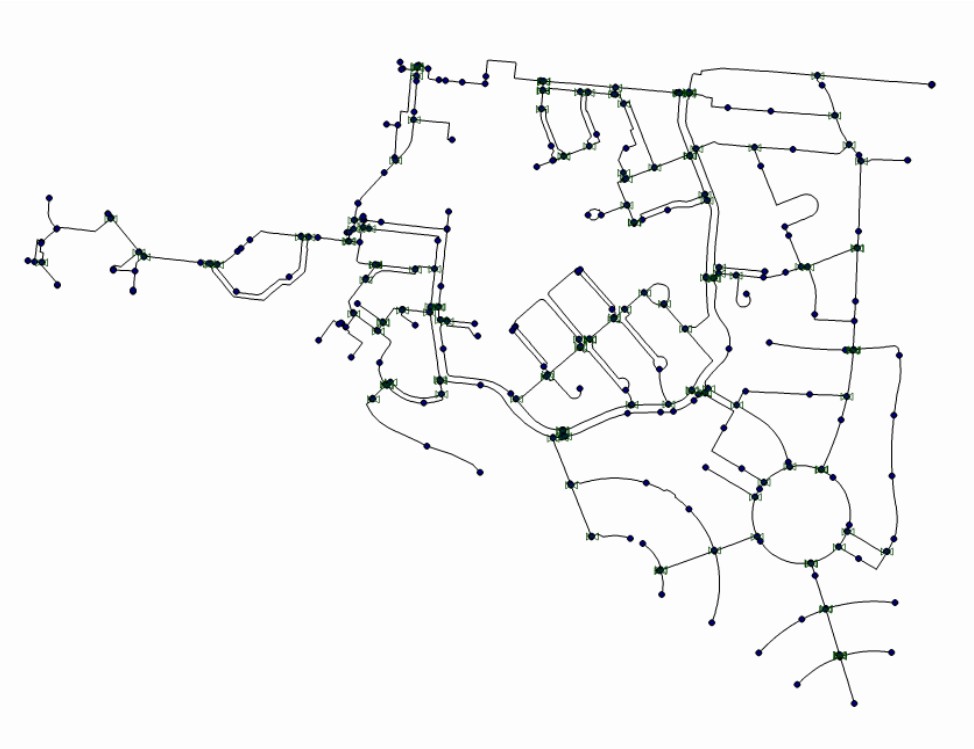

**Figure 1: EPANET model of the drinking water distribution network of Sittard (Netherlands), consisting of 584 nodes, 491 links and 1 reservoir.**

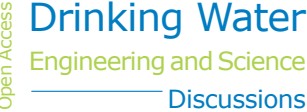



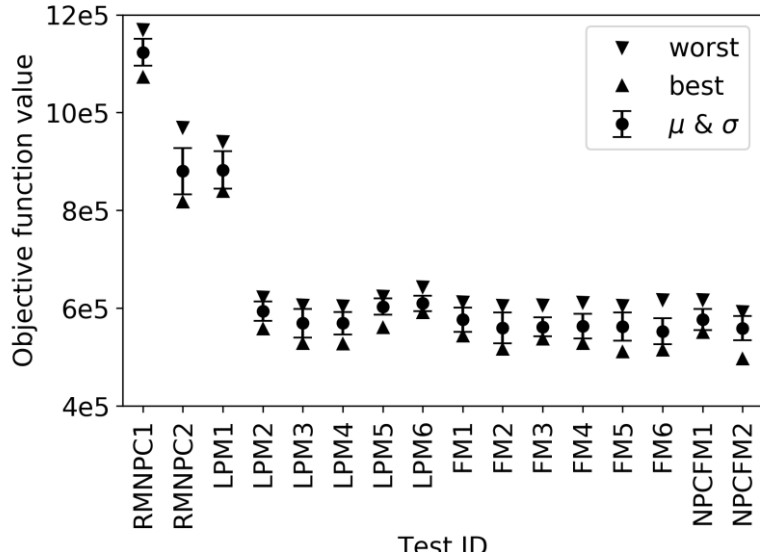

**Figure 2. Overview of the obtained results for the different tests.**

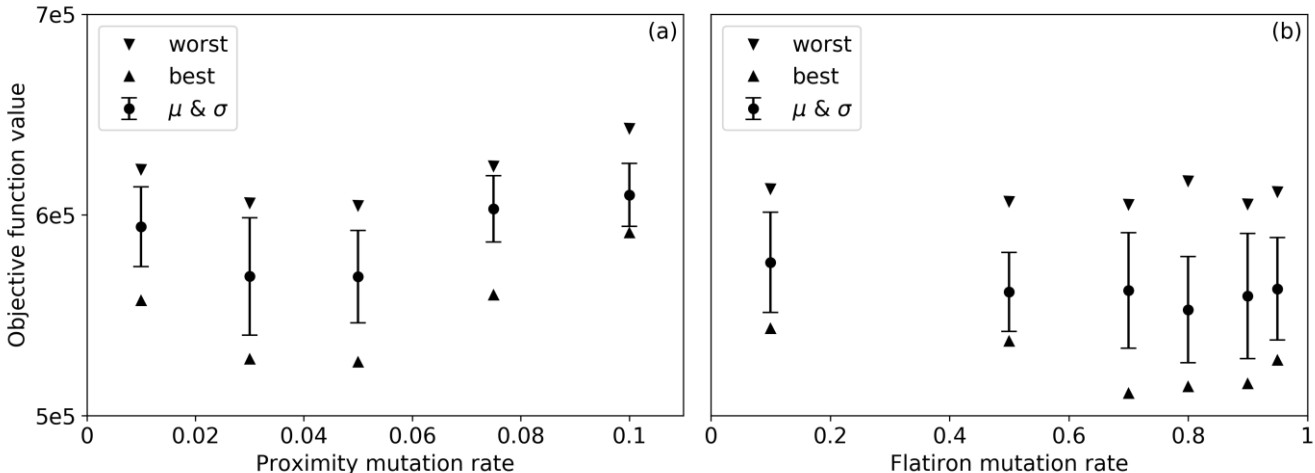

**Figure 3. (a) Influence of the proximity mutation rate on the obtained objective function values in tests LPM2, LPM3, LPM4, LPM5 and LPM6. (b) Influence of the flatiron mutation rate on the obtained objective function values in tests FM1, FM2, FM3, FM4, FM5 and FM6.**





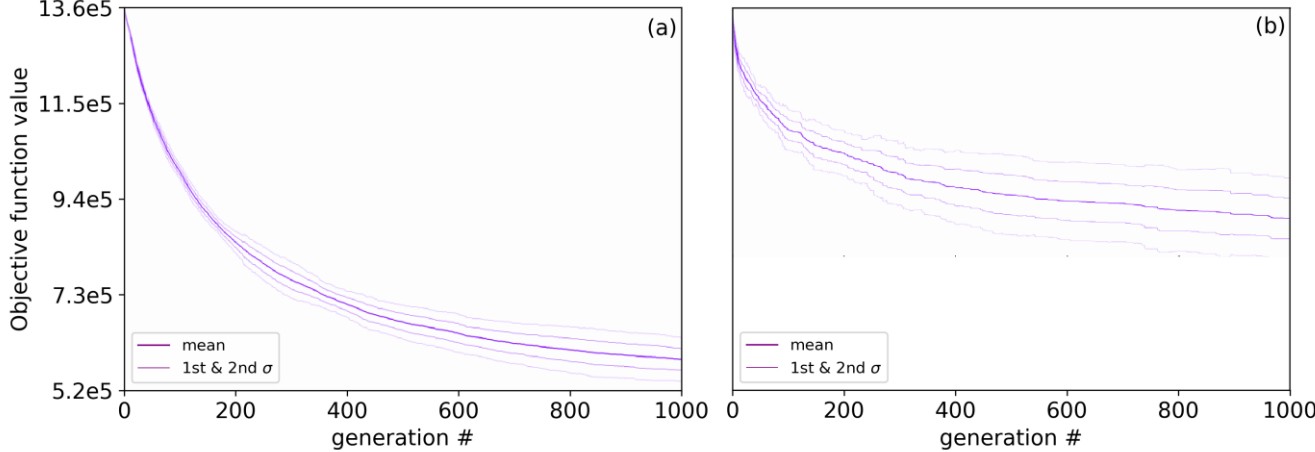

**Figure 4. Convergence curves (mean, first and second standard deviations of 10 runs) obtained for tests 2 and 12. (a) Random mutation and one-point crossover (test 2). (b) One-point crossover, proximity mutation and flatiron mutation (test 12).**

**Table 1. Specific variators. Types include mutators (m) and crossover (c), classic (C) and heuristic (H).**

| variator | acronym | type | description |
|---|---|---|---|
| random mutation | RM | mC | assign a random value within a prescribed range to a parameter or subdivision index |
| n point crossover | NPC | cC | mix lists of decision variable (attribute/parameter) values |
| selection mutation | SeM | mC | random selection from a list of predefined values |
| flatiron mutation | FM | mH | give an object the same value as (or the minimum or maximum of) its neighbours on both sides, provided each side has only one neighbour |
| list proximity mutation | LPM | mH | random selection from $n$ nearest neighbour values in an ordered list of allowed values |

**Table 2. Available pipe diameters (mm).**

| 0 | 13.2 | 21.2 | 36 | 42.6 | 58.2 | 66 | 72.8 |
|---|---|---|---|---|---|---|---|
| 87.3 | 101.6 | 130.8 | 147.6 | 163.6 | 190 | 200 | |

**Table 3. Problem specific variator values considered in the different tests and obtained results for 10 runs with 1x10e5 function evaluations each. The best results are indicated in bold.**

| Test | | Crossover rate (N) | Random Mutation (N) | Proximity mutation (H) | Flatiron mutation (H) | Mean | Std. | Best | Worst |
|---|---|---|---|---|---|---|---|---|---|
| 1 | RM-NPC1 | 0.9 | 0.1 | 0 | 0 | 1.12E+06 | 2.76E+04 | 1.07E+06 | 1.17E+06 |
| 2 | RM-NPC2 | 0.95 | 0.05 | 0 | 0 | 8.80E+05 | 4.75E+04 | 8.18E+05 | 9.70E+05 |

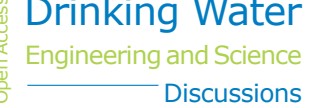

| | | | | | | | | | |
|---|---|---|---|---|---|---|---|---|---|
| 3 | LPM1 | 0.95 | 0.05 | 0.01 | 0 | 8.83E+05 | 3.83E+04 | 8.39E+05 | 9.41E+05 |
| 4 | LPM2 | 0.95 | 0 | 0.01 | 0 | 5.94E+05 | 1.99E+04 | 5.57E+05 | 6.23E+05 |
| 5 | LPM3 | 0.95 | 0 | 0.03 | 0 | 5.69E+05 | 2.92E+04 | 5.28E+05 | 6.06E+05 |
| 6 (5) | LPM4 | 0.95 | 0 | 0.05 | 0 | 5.69E+05 | 2.30E+04 | 5.27E+05 | 6.05E+05 |
| 7 | LPM5 | 0.95 | 0 | 0.075 | 0 | 6.03E+05 | 1.65E+04 | 5.60E+05 | 6.24E+05 |
| 8 6 | LPM6 | 0.95 | 0 | 0.1 | 0 | 6.10E+05 | **1.57E+04** | 5.91E+05 | 6.43E+05 |
| 9 7 | FM1 | 0.95 | 0 | 0.05 | 0.1 | 5.76E+05 | 2.49E+04 | 5.43E+05 | 6.13E+05 |
| 10 8 | FM2 | 0.95 | 0 | 0.05 | 0.9 | 5.60E+05 | 3.11E+04 | 5.16E+05 | 6.05E+05 |
| 11 9 | FM3 | 0.95 | 0 | 0.05 | 0.5 | 5.62E+05 | 1.97E+04 | 5.37E+05 | 6.07E+05 |
| 12 10 | FM4 | 0.95 | 0 | 0.05 | 0.95 | 5.63E+05 | 2.55E+04 | 5.28E+05 | 6.12E+05 |
| 13 11 | FM5 | 0.95 | 0 | 0.05 | 0.7 | 5.62E+05 | 2.88E+04 | **5.11E+05** | 6.05E+05 |
| 14 12 | FM6 | 0.95 | 0 | 0.05 | 0.8 | **5.53E+05** | 2.64E+04 | 5.14E+05 | 6.17E+05 |
| 15 13 | NPC-FM1 | 0.9 | 0 | 0.05 | 0.8 | 5.77E+05 | 2.16E+04 | 5.50E+05 | 6.17E+05 |
| 16 14 | NPC-FM2 | 0.8 | 0 | 0.05 | 0.8 | 5.59E+05 | 2.46E+04 | 4.97E+05 | **5.93E+05** |