# Peer review of "Technical note: Problem specific variators in a genetic algorithm for the optimization of drinking water networks"

_Drinking Water Engineering and Science, 2018_

## Referee Comment (RC1) · Anonymous Referee #1 · 23 Sep 2018

The authors have done a fairly extensive study of genetic algorithm operators for a drinking water network optimization problem. This is important work, especially as optimization of this type becomes more prevalent in engineering practice. I found the paper to be well written and concise, but a few minor issues should be addressed:

1. page 1, line 16 - "A widely applied approach is that of Genetic Algorithms..." the authors should cite the Maier et al (2014) study here (references to it do appear later in the paper). This Maier paper is a thorough review of the state of the field, so it can be used to contextualize the work. In fact, I would say that modern terminology calls these approaches "evolutionary algorithms", of which the generic Genetic Algorithm is

simply one approach.

2. page 1, line 18 - If the authors are referring to "tricks" they should cite a few examples. Otherwise, the authors are giving the false impression that they are the first to explore these genetic algorithm parameters, which I do not believe they are.

3. page 2 - In this list, I think it is important to mention the fact that some of these evolutionary algorithm techniques are not genetic algorithms at all, but rather other algorithms such as differential evolution. THat could perhaps be a sixth item in the list.

4. section "Problem specific variators" - I was somewhat confused by this section, since in it the authors have discussed some of the new operators they are looking at, but it seemed to overlap with the subsequent "Tests" section. I think the organization of these methods sections could use some fixing.

5. A citation for EPANET should be given, and if the model is freely available, that should be noted.

6. page 3, line 12 - The authors have consistently forgotten the formatting of 10ˆ5; this should be fixed throughout.

7. A brief mention of how the optimization was carried out would be appropriate, especially in the population size. The population size has been shown to be a very important parameter for assessing GA success.

8. The "Tests" section should also briefly comment on the performance criterion used (namely, the best optimal solution found so far as a function of time during the run)

9. The authors should give some general guidance as to how to assess the "significance" of the objective function value. For example, other than the drastic reduction in objective of the first three tests as compared to the last three, is there a difference? In other words, it appears as though the first three tests are the worst, but can we tell the difference with the other 9 tests?

---

## Short Comment (SC1) · 28 Sep 2018

Dear reviewer, First of all many thanks for your suggestions. Following you comment on different types of evolutionary algorithms, we learned that the term 'genetic algorithms' seems to be specifically tied to the use of a binary representation of the problem. If that is the case, our approach is not a GA at all, since we use a natural discription of the problem and use problem specific operators. Accordingly, we intend to forsake the term GA throughout the paper in favour of EA. Would you consider this appropriate? Best regards, Karel
* * *
[Figure]

21, 2018.

---

## Referee Comment (RC2) · Anonymous Referee #2 · 1 Oct 2018

Authors present a set of different genetic operators that may improve the behaviour of the algorithm in terms of convergence and repetitiveness. The paper begins with a fast review of the state of the art, stating the areas where these algorithms might be improved. Then, authors present the variations they proposed for the different genetic operators. Finally, a set of simulations were done to test the performance of the operators. In this sense, the paper can improved the standard use of Genetic Algorithm in the field of the design of water networks. There are some minor comments that could be mentioned:

a) Authors propose the use of two heuristic variations for mutation operator: flatiron

and proximity. A doubt exists about their authorship. Do the authors propose these operators? Are they adapted from previous works? In the first case, the operators should be described with more detail. On the contrary, if the operators were taken from other authors, their work should be referenced.

b) Proximity operator use values of a number of neighbours. The concept of neighbour in water system is a topological one. Topological operations might be computational heavy and time to reach a solution might increase. Some comment about this fact could be said.

c) Authors made a set of tests with "[...] 1x10ˆ5 function evaluation [...]". Later, in Figure 4 up to 1000 generations are represented. It can be deduced that the population used for the algorithm is 100 individuals. However, since Genetic Algorithm is strongly based on population, the population must be explicitly mentioned in the paper.

Apart from the previous comments, other little changes could help improving the paper:

- In line 1-2, page 2, the year of publication for El-Mihoub et al. (2006) might be added.

- In line 13, page 2, there is a mistake: "[...] to construct an model [...]" should be changed by "[...] to construct a model [...]"

- All references to the number of function evaluations (1x105) or fitness values (8.8x105) should be written with superscript or using the "ˆ" symbol. This affects to lines 12, 22, 24 and 29 in page 3; and lines 9 and 12 in page 4.

- In line 6, page 3 there is a word missing: "[...] 584 nodes, 491 ? and 1 reservoir [ ...]". I guess the word links or pipes is missing. Anyway, please check this number since a network with 584 nodes and (only) 491 links cannot be run with EPANET. If this check is performed, Figure 4 caption should also be corrected.

- In line 2, page 4 the test 2 is referred to as RM-NPC2. However, in Figure 2 is referred to as RMNPC2 and in Figure 4 is referred to simply as test 2. Please use the same reference throughout the paper.

- Caption for Figure 4 refers part (a) as results corresponding to test 2 and part (b) as results for test 12. However, the part (a) has a value for the objective function between 5.2e5 and 7.3e5 (coherent with the results of test 12, 5.63e5). It happens the same with part (b), values close to 9e5, more likely to correspond to test 2, 8.8e5. Probably it is just a question of flipping the caption for parts (a) and (b).

- (This is only my personal point of view. Please forget if you disagree). In scientific texts, it is recommended to use a non-personal language. For that reason I encourage the authors to change the personal style in line 22, page 1 ("[. . .] In this paper, we [. . .]") and line 13, page 4 ("[. . .] consulting projects, we will [. . .]").

---

## Author Comment (AC1) · 11 Oct 2018

**Reviewer Comments:**

**Reviewer 1**

A. page 1, line 16 - "A widely applied approach is that of Genetic Algorithms..." the authors should cite the Maier et al (2014) study here (references to it do appear later in the paper). This Maier paper is a thorough review of the state of the field, so it can be used to contextualize the work. In fact, I would say that modern terminology calls these approaches "evolutionary algorithms", of which the generic Genetic Algorithm is simply one approach.

B. page 1, line 18 - If the authors are referring to "tricks" they should cite a few examples. Otherwise, the authors are giving the false impression that they are the first to explore these genetic algorithm parameters, which I do not believe they are.

C. page 2 - In this list, I think it is important to mention the fact that some of these evolutionary algorithm techniques are not genetic algorithms at all, but rather other algorithms such as differential evolution. THat could perhaps be a sixth item in the list.

D. section "Problem specific variators" - I was somewhat confused by this section, since in it the authors have discussed some of the new operators they are looking at, but it seemed to overlap with the subsequent "Tests" section. I think the organization of these methods sections could use some fixing.

E. A citation for EPANET should be given, and if the model is freely available, that should be noted.

F. page 3, line 12 - The authors have consistently forgotten the formatting of 10ˆ5; this should be fixed throughout.

G. A brief mention of how the optimization was carried out would be appropriate, especially in the population size. The population size has been shown to be a very important parameter for assessing GA success.

H. The "Tests" section should also briefly comment on the performance criterion used (namely, the best optimal solution found so far as a function of time during the run)

I. The authors should give some general guidance as to how to assess the "significance" of the objective function value. For example, other than the drastic reduction in objective of the first three tests as compared to the last three, is there a difference? In other words, it appears as though the first three tests are the worst, but can we tell the difference with the other 9 tests?

**Reviewer 2**

J. Authors propose the use of two heuristic variations for mutation operator: flatiron and proximity. A doubt exists about their authorship. Do the authors propose these operators? Are they adapted from previous works? In the first case, the operators should be described with more detail. On the contrary, if the operators were taken from other authors, their work should be referenced.

K. Proximity operator use values of a number of neighbours. The concept of neighbour in water system is a topological one. Topological operations might be computational heavy and time to reach a solution might increase. Some comment about this fact could be said.

L. Authors made a set of tests with "[: : :] 1x10ˆ5 function evaluation [: : :]". Later, in Figure 4 up to 1000 generations are represented. It can be deduced that the population used for the algorithm is 100 individuals. However, since Genetic Algorithm is strongly based on population, the population must be explicitly mentioned in the paper.

M. Apart from the previous comments, other little changes could help improving the paper:

   - In line 1-2, page 2, the year of publication for El-Mihoub et al. (2006) might be added.

   - In line 13, page 2, there is a mistake: "[: : :] to construct an model [: : :]" should be changed by "[: : :] to construct a model [: : :]"

   - All references to the number of function evaluations (1x105) or fitness values (8.8x105) should be written with superscript or using the "ˆ" symbol. This affects to lines 12, 22, 24 and 29 in page 3; and lines 9 and 12 in page 4.

   - In line 6, page 3 there is a word missing: "[: : :] 584 nodes, 491 ? and 1 reservoir [ : : :]". I guess the word links or pipes is missing. Anyway, please check this number since a network with 584 nodes and (only) 491 links cannot be run with EPANET. If this check is performed, Figure 4 caption should also be corrected.

   - In line 2, page 4 the test 2 is referred to as RM-NPC2. However, in Figure 2 is referred to as RMNPC2 and in Figure 4 is referred to simply as test 2. Please use the same reference throughout the paper.

   - Caption for Figure 4 refers part (a) as results corresponding to test 2 and part (b) as results for test 12. However, the part (a) has a value for the objective function between 5.2e5 and 7.3e5 (coherent with the results of test 12, 5.63e5). It happens the same with part (b), values close to 9e5, more likely to correspond to test 2, 8.8e5. Probably it is just a question of flipping the caption for parts (a) and (b).

   - (This is only my personal point of view. Please forget if you disagree). In scientific texts, it is recommended to use a non-personal language. For that reason I encourage the authors to change the personal style in line 22, page 1 ("[: : :] In this paper, we [: : :]") and line 13, page 4 ("[: : :] consulting projects, we will [: : :]").

**Author responses:**

We thank the reviewers for their thorough suggestions. The responses to the individual point follow below.

To reviewer 1:

A. We have added the reference to Maier to this point in the text and modified the text to include the broader term of evolutionary algorithms.

B. We have added the reference to El-Mihoub's review of HGA variants to this point of the text. We have also rephrased some parts of the introduction to avoid giving the impression of claiming to be the first to investigate this.

C. We have added this insight that the hybridization changes the GA's place in the EA taxonomy to the end of the list.

D. We have restructured the methods section accordingly.

E. We have added the reference. The network model is property of a Dutch utility and is not freely available.

F. We have reformatted to $10^5$ accordingly.

G. We have added the various settings of the algorithm to the relevant part of the methods section.

H. We now explicitly note the performance criterion in the same section.

I. We have added some thoughts about this to the discussion.

To reviewer 2:

J. Now describe both variators in some more detail, and have rephrased to stress that only one of them is custom made.

K. We have added a comment on the retrieval of neighbours. (This was added to the description of the flatiron mutator, which uses neighbours, whereas the proximity mutator does not).

L. We have added this information to the methods section.

M. We agree with the reviewer on all items mentioned, and have modified the text accordingly.

**Revised Manuscript with track changes:**

[revised manuscript text omitted]

1. for the mutating pipe, obtain the neighbour IDs from a lookup table with neighbouring pipes per pipe (it is worth noting that this lookup table is created at the start of the optimization, thereby limiting its impact on computation);
2. if the pipe connects to exactly 1 or 2 neighbouring pipes, compare the diameter of the mutating pipe to those of its neighbours;
3. if the mutating diameter is larger than the diameter of all neighbours, reduce it to the largest diameter among neighbours.

The heuristic list proximity mutator enhances convergence according to approach 3 in the list above. It is equivalent to the classic 'creep mutator' (Sivanandam & Deepa, 2007): it functions as the regular random mutation of a single pipe diameter, except that the possible outcomes of the mutation are limited to values close to the value prior to mutation. . It limits the possible outcomes of a mutation to diameters close to the original value, This mutator is typically used because large deviations from the original diameter are likely to cause hydraulically inviable solutions.

[revised manuscript text omitted]

**5 Discussion and conclusions**

The results presented in this paper clearly illustrate the value of applying heuristic, non-classical variators in drinking water distribution system design optimizations using genetic algorithms. While the difference between the test with random mutation and the other tests is especially noticeable in Fig. 2, it is worth noting that the smaller differences between the other individual tests indicate a significant difference in convergence as well. In Fig 4a, for instance, it can be seen that, in FM4, the average objective function value of $6x10^5$ was reached in around 700 generations, about 1.4 times faster than in LPM2, LPM5 and LMP6.

In our the tests, the combination of a low rate for the proximity mutation with a high rate of the flatiron mutation leads to the best results after $1x10^5$ function evaluations (test numbers 13:FM5 and 14:FM6), i.e. the fastest convergence. All tested combinations which include either the flatiron or the proximity mutation exhibit a similar or worse performance. Albeit slower, particularly stable results were obtained with the proximity mutation (rate=0.1) and no flatiron mutation. These runs show the smallest standard deviation in the results after $1x10^5$ function evaluations.

In our future research and consulting projects with Gondwana, we will continue to use thisthis combination of variators will be used in order to deal with the computational challenges of larger real world networks.

**Acknowledgments**

The authors wish to thank Henk Vogelaar from Waterleiding Maatschappij Limburg (WML) for providing the Sittard network model used in the calculations.

[Figure]

**Figure 1: EPANET model of the drinking water distribution network of Sittard (Netherlands), consisting of 584 583 nodesjunctions,**
15 **491 links pipes, 140 valves and 1 reservoir.**

[Figure]

**Figure 2. Overview of the obtained results for the different tests.**

[Figure]

**Figure 3. (a) Influence of the proximity mutation rate on the obtained objective function values in tests LPM2, LPM3, LPM4, LPM5**
5   **and LPM6. (b) Influence of the flatiron mutation rate on the obtained objective function values in tests FM1, FM2, FM3, FM4, FM5**
**and FM6.**

[Figure]

**Figure 4. Convergence curves (mean, first and second standard deviations of 10 runs) obtained for tests**  **RMNPC2 and** **FM4. (a) One-point crossover, proximity mutation and flatiron mutation (FM4). (a) Random mutation and one-point crossover (RMNPC2). (b) One-point crossover, proximity mutation and flatiron mutation (test 12).**

**Table 1. Specific variators. Types include mutators (m) and crossover (c), classic (C) and heuristic (H).**

| variator | acronym | type | description |
|---|---|---|---|
| random mutation | RM | mC | assign a random value within a prescribed range to a parameter or subdivision index |
| n point crossover | NPC | cC | mix lists of decision variable (attribute/parameter) values |
| selection mutation | SeM | mC | random selection from a list of predefined values |
| flatiron mutation | FM | mH | give an object the same value as (or the minimum or maximum of) its neighbours on both sides, provided each side has only one neighbour |
| list proximity mutation | LPM | mH | random selection from $n$ nearest neighbour values in an ordered list of allowed values |

**Table 2. Available pipe diameters (mm).**

| 0 | 13.2 | 21.2 | 36 | 42.6 | 58.2 | 66 | 72.8 |
|---|---|---|---|---|---|---|---|
| 87.3 | 101.6 | 130.8 | 147.6 | 163.6 | 190 | 200 | |

**Table 3. Problem specific variator values considered in the different tests and obtained results for 10 runs with 1x10e5 function evaluations each. The best results are indicated in bold.**

| Test | Crossover rate (N) | Random Mutation (N) | Proximity mutation (H) | Flatiron mutation (H) | Mean | Std. | Best | Worst |
|---|---|---|---|---|---|---|---|---|
| 1  RM-NPC1 | 0.9 | 0.1 | 0 | 0 | 1.12E+06 | 2.76E+04 | 1.07E+06 | 1.17E+06 |

| 2 | RM–NPC2 | 0.95 | 0.05 | 0 | 0 | 8.80E+05 | 4.75E+04 | 8.18E+05 | 9.70E+05 |
| 3 | LPM1 | 0.95 | 0.05 | 0.01 | 0 | 8.83E+05 | 3.83E+04 | 8.39E+05 | 9.41E+05 |
| 4 | LPM2 | 0.95 | 0 | 0.01 | 0 | 5.94E+05 | 1.99E+04 | 5.57E+05 | 6.23E+05 |
| 5 | LPM3 | 0.95 | 0 | 0.03 | 0 | 5.69E+05 | 2.92E+04 | 5.28E+05 | 6.06E+05 |
| 6  | LPM4 | 0.95 | 0 | 0.05 | 0 | 5.69E+05 | 2.30E+04 | 5.27E+05 | 6.05E+05 |
| 7 | LPM5 | 0.95 | 0 | 0.075 | 0 | 6.03E+05 | 1.65E+04 | 5.60E+05 | 6.24E+05 |
| 8  | LPM6 | 0.95 | 0 | 0.1 | 0 | 6.10E+05 | **1.57E+04** | 5.91E+05 | 6.43E+05 |
| 9  | FM1 | 0.95 | 0 | 0.05 | 0.1 | 5.76E+05 | 2.49E+04 | 5.43E+05 | 6.13E+05 |
| 10  | FM2 | 0.95 | 0 | 0.05 | 0.9 | 5.60E+05 | 3.11E+04 | 5.16E+05 | 6.05E+05 |
| 11  | FM3 | 0.95 | 0 | 0.05 | 0.5 | 5.62E+05 | 1.97E+04 | 5.37E+05 | 6.07E+05 |
| 12  | FM4 | 0.95 | 0 | 0.05 | 0.95 | 5.63E+05 | 2.55E+04 | 5.28E+05 | 6.12E+05 |
| 13  | FM5 | 0.95 | 0 | 0.05 | 0.7 | 5.62E+05 | 2.88E+04 | **5.11E+05** | 6.05E+05 |
| 14  | FM6 | 0.95 | 0 | 0.05 | 0.8 | **5.53E+05** | 2.64E+04 | 5.14E+05 | 6.17E+05 |
| 15  | NPC–FM1 | 0.9 | 0 | 0.05 | 0.8 | 5.77E+05 | 2.16E+04 | 5.50E+05 | 6.17E+05 |
| 16  | NPC–FM2 | 0.8 | 0 | 0.05 | 0.8 | 5.59E+05 | 2.46E+04 | 4.97E+05 | **5.93E+05** |